# A Review of 3D Printing Technology in Pharmaceutics: Technology and Applications, Now and Future

**DOI:** 10.3390/pharmaceutics15020416

**Published:** 2023-01-26

**Authors:** Shanshan Wang, Xuejun Chen, Xiaolu Han, Xiaoxuan Hong, Xiang Li, Hui Zhang, Meng Li, Zengming Wang, Aiping Zheng

**Affiliations:** 1College of Biotechnology, Tianjin University of Science and Technology, Tianjin 300457, China; 2State Key Laboratory of Toxicology and Medical Countermeasures, Beijing Institute of Pharmacology and Toxicology, Beijing 100850, China

**Keywords:** three-dimensional printing technology, three-dimensional printed drug, drug delivery, personalized medicine, research status

## Abstract

Three-dimensional printing technology, also called additive manufacturing technology, is used to prepare personalized 3D-printed drugs through computer-aided model design. In recent years, the use of 3D printing technology in the pharmaceutical field has become increasingly sophisticated. In addition to the successful commercialization of Spritam^®^ in 2015, there has been a succession of Triastek’s 3D-printed drug applications that have received investigational new drug (IND) approval from the Food and Drug Administration (FDA). Compared with traditional drug preparation processes, 3D printing technology has significant advantages in personalized drug manufacturing, allowing easy manufacturing of preparations with complex structures or drug release behaviors and rapid manufacturing of small batches of drugs. This review summaries the mechanisms of the most commonly used 3D printing technologies, describes their characteristics, advantages, disadvantages, and applications in the pharmaceutical industry, analyzes the progress of global commercialization of 3D printed drugs and their problems and challenges, reflects the development trends of the 3D printed drug industry, and guides researchers engaged in 3D printed drugs.

## 1. Introduction

In contrast to the traditional manufacturing techniques of “subtractive manufacturing”, 3D printing is an “additive manufacturing” technology, where a model is constructed using computer-aided design software, sliced, and transferred to a printer, and the 3D product is then constructed layer by layer using the principle of layered manufacturing [1,2]. With the research and development of 3D printing technology, many new 3D printing technologies have emerged one after another. As each 3D printing technology uses different materials, deposition techniques, layering manufacturing mechanisms, and final product characteristics, the American Society for Testing and Materials classified 3D printing technologies into seven categories according to their technical principles [3,4], namely material extrusion, binder jetting, powder bed fusion, vat photopolymerization, material jetting, directed energy deposition, and sheet lamination.

Three-dimensional printing technology is widely used in automotive, construction, aerospace, medical, and many other fields. In the pharmaceutical sector, research into 3D printing technology is currently experiencing a global boom [5,6]. Compared to traditional preparation technologies, 3D printing offers flexibility in the design of complex 3D structures within drugs, the adjustment of drug doses and combinations, and rapid manufacturing and prototyping, enabling precise control of drug release to meet a wide range of clinical needs, a high degree of flexibility and creativity to personalize pharmaceuticals, and a significant reduction in preparation development time, driving a breakthrough in drug manufacturing technology and transforming the way we design, manufacture, and use drugs [7,8,9]. Three-dimensional printing technologies have been used to manufacture a variety of medicinal products, such as immediate-release tablets, controlled-release tablets, dispersible films, microneedles, implants, and transdermal patches [10]. The main 3D printing technologies used in pharmaceuticals are BJ-3DP, FDM, SSE, and MED in material extrusion, and SLA [11]. Table 1 describes the characteristics of these technologies at each stage of drug preparation and assesses the advantages and disadvantages of each technology.

This review aims to provide a multidimensional analysis of several commonly used representative 3D printing technologies, introduce the current status of their application and manufacturing principles in the field of pharmaceutics, and clarify the advantages and disadvantages of each technology and the suitable pharmaceutical dosage forms to be manufactured. At the same time, combined with the literature, we sort out and review the current status of the industrialization of 3D printing technology in the pharmaceutical field and the problems or challenges it faces.

## 2. The Advantages of 3D Printing Technology in Pharmaceuticals

### 2.1. Personalized Medicine for Special Populations

The health and safety of medication for special populations such as the elderly and children has long been an issue of concern [12,13]. Children are in a period of growth and development and have a particular reactivity and sensitivity to medication; the elderly have a reduced absorption and metabolism capacity, and the coexistence of multiple diseases and combined medication is very common [14]. Whereas current drug dosages are standardized, there are few specialized drugs for special populations, and children’s medication is often administered by manually breaking tablets, which is not only inaccurate but may also damage the particular structure of the preparation and cause adverse reactions [15].

Three-dimensional printing technology is highly flexible and can be used to print targeted medicines by adjusting model parameters such as size, shape, or fill rate [16]. For pediatric patients, 3D printing technology can be used to produce low-dose personalized medicines suitable for children, and can also be used to improve the appearance and taste of the medicines to increase the compliance of pediatric patients [17,18,19]; for elderly patients who have difficulty swallowing, 3D printing technology can prepare loose and porous preparations, thus, helping them to take medication; for patients who take multiple drugs at the same time, different drugs can be partitioned and combined into a single tablet to avoid errors or missed drugs, which can increase the safety and effectiveness of medication; in addition, specially shaped preparations can be printed or special symbols can be printed on the surface of the preparation to provide convenience for patients with visual impairment [20,21,22]. The advantages of 3D printing technology for personalized drug delivery provide technical support for people to achieve personalized medicine, and some 3D printed drug companies are moving towards the goal of personalized medicine, such as FabRx in the UK, which prepares personalized drugs for children with maple diabetes, and has placed SSE printers in the pharmacy of a Spanish hospital and conducted clinical trials on the subject [23].

### 2.2. Precise Control of Drug Release

As the most widely used solid oral dosage form, tablets account for 70% of all dosage form production [24]. Traditional manufacturing processes enable tablets to be produced at a lower cost, but they have been less creative in preparation development, with long development times and less ability to manufacture personalized preparations on demand.

Compared to conventional tablets, controlled-release preparations allow for precise control of drug release, avoiding side effects and improving efficacy. However, traditional manufacturing processes pose greater challenges in the development and manufacture of controlled-release preparations due to their limitations. Three-dimensional printing technology is highly flexible and is well suited to the development and manufacture of complex preparations through the combination of different drugs, the design of complex models, and the adjustment of printing parameters [25].

For example, Triastek’s 3D printed product, T19, which received IND approval from the FDA in January 2021, is a controlled release preparation designed for the circadian rhythm of rheumatoid arthritis, where patients take it at bedtime and blood concentration peaks in the morning with the most severe symptoms of pain, joint stiffness, and dysfunction, and maintains daytime blood concentration for optimal therapeutic effect, providing better medication options for patients [26].

### 2.3. Rapid Integration of Production

In the large-scale production of drugs, ordinary pharmaceutical companies, to meet the global demand for traditional drugs, usually have a very high production capacity, and their production equipment is usually large, with a relatively single type of equipment, lacking the necessary production flexibility to quickly complete the cleaning and change the variety of drugs produced. Three-dimensional printing technology, on the other hand, can integrate rapid manufacturing, with compact equipment, fewer production steps, automated and digital production processes, and the ease of changing the variety of drugs produced. For example, SSE technology allows for the direct replacement of disposable syringes containing different drug varieties to meet the needs of multiproduct production equipment [27].

Furthermore, in the drug development phase, 3D printing technology is well suited to small-scale drug production that requires customization and frequent design modifications due to its lower small-batch production costs and integrated manufacturing process, which can play an important role in conditions of limited time and resources. This has very important implications for drug development, with Merck using 3D printing technology to accelerate clinical trials and predicting through data that in clinical phases I–III, preparation development time will be reduced by 60% and the API required to prepare the medication will be reduced by 50% [28].

## 3. Principle of BJ-3DP Technology and Applications in the Pharmaceutical Industry

### 3.1. The Principle of BJ-3DP Technology

BJ-3DP is the primary 3D printing technology used for drug production [29]. The printing principle is shown in Figure 1A. First, the roller spreads a thin powder layer on the platform, the droplets are sprayed through the removable printhead, and they selectively bind the powder together; then the platform is lowered, the roller spreads a new powder layer, the print head continues to add droplets, using the principle of layer-by-layer printing, and so on until complete; finally, the preparations are removed, the adhering powder is removed, and postprocessing is carried out [30]. Printing inks can contain only the binder, while the powder bed contains the API and other excipients. API can also be sprayed into the powder bed as a solution or as a suspension of nanoparticles [31]. The APIs that can be used for the BJ-3DP technology are not only those with good water solubility, but for insoluble APIs, the solubility of the APIs can also be improved by means of pretreatment, but there are fewer relevant studies. For example, Kozakiewicz-Latała et al. [32] used the hydrophobic API clotrimazole as a model drug and prepared a suspension with the hydrophilic excipients PVP and lactose in a certain ratio, followed by spray drying, which increased the wettability and printability of clotrimazole.

The mechanism of the BJ-3DP technology is complex, and the printing process for this technology can be roughly divided into three steps: (1) droplet formation, (2) selective binding of droplets to powder, and (3) drying or curing of the finished product [33].

Droplet formation is a complex process involving the formation and elongation of filaments, the necking, breaking, and rebounding of filaments, and the formation and fusion of primary and satellite droplets, as shown in Figure 1B. Important physical parameters affecting the properties of printing inks include viscosity, density, and surface tension, which affect the droplet formation mechanism and droplet volume and velocity. Reis and Derby [34] used computational fluid dynamics to model the free surface flow characteristics of droplet formation based on Fromm’s prediction of the range of dimensionless Z values representing the printability of the ink for the formation of stable droplets of printing ink [35], and explored the effect of fluid properties on droplet ejection in conjunction with parallel experiments to determine that Z values should be between 1 and 10. Jang et al. [36] investigated the ejection effect of inks consisting of ethanol, water, and glycol, examining the dynamic process of droplet ejection, leading to a redefinition of the Z value range from 4 to 14. To obtain the best droplet ejection quality, the droplets generated by the printhead should preferably be in the form of monodisperse droplets, that is, only a single droplet is generated per pulse cycle.

When droplets impact on a smooth, nonporous surface, droplet diffusion depends mainly on droplet volume and equilibrium contact angle [37,38]. However, the effect of droplets impacting powder beds is much more complex, as shown in Figure 1C. Yarin’s study found that the initial impact phase was controlled by kinematic behavior, mainly by inertial forces, followed by droplet diffusion, recoil, and oscillation driven by impact, and later capillary forces dominated and controlled the diffusion process [39]. The effect of droplets impacting on powder beds has been studied, and researchers are continuously investigating the relationship between the dimensionless number of ejected droplets and the effect of droplets impinging on a powder bed [40,41].

Finally, the drying or curing process can also affect the quality of the final product. In most cases, drying takes place by evaporation of the solvent, so the evaporation rate is an important parameter in the choice of solvent. For example, using polymer-API-solvent systems as printing inks to prepare amorphous solid dispersions after the droplets have dried [42,43,44], provides an effective means of preparing low-dose drugs from insoluble APIs. In addition, binder concentration, nozzle diameter, droplet spacing, print speed, and the frequency and speed of droplet generation are all factors that should be considered during the printing process [45].

### 3.2. BJ-3DP Technology in Pharmaceuticals

The first paper on the use of BJ-3DP in pharmaceuticals was published in 1996 [46], demonstrating the feasibility of using 3D printing technology to manufacture medication, and since then research has been carried out using BJ-3DP technology to prepare various dosage forms, such as immediate release, slow and controlled release, and compounded and implant preparations [47,48]. Table 2 details the various preparations that have been performed using BJ-3DP technology in recent years. In the published studies, BJ-3DP technology has been used in two main categories of preparations, namely oral solid dosage forms and subcutaneous implants.

As a new technology for the preparation of oral solid dosage forms, early research focused on the validation of the feasibility of complex preparations. When droplets are selectively deposited onto the powder bed, the preparation of multilayer compounded preparations is only possible with the integral replacement of the powder bed material due to the homogeneous composition of the powder bed. While the print head can be loaded with inks of different compositions, parameters such as droplet volume or velocity, number of ejections, and deposition position can be adjusted to enable the preparation of complex preparations. Spritam^®^, the first 3D-printed preparation to be launched in 2015, was prepared by BJ-3DP technology and is a dispersible tablet with a high drug-loading capacity without complex structures [49]. This preparation method reflects the technical characteristics of the BJ-3DP, which is to prepare tablets solely through the contact adhesion of the powder and ink, thus, generating a porous structure that disintegrates rapidly, unlike the mechanical forces of conventional technology. As research into BJ-3DP technology intensifies, the types of preparations developed by BJ-3DP technology now focus on immediate-release preparations.

In addition, scaffolds prepared by BJ-3DP can mimic natural bone and show great potential in the preparation of bone scaffolds, while having a loose and porous structure and high surface roughness [50,51], which is very conducive to cell attachment and growth; Table 2 lists the relevant studies in recent years. Compared to traditional processes, 3D printing technology allows better control of factors such as the shape, size, and internal structure of the scaffold, allowing the implant to fit the delivery site to the greatest extent possible.

**Table 2 pharmaceutics-15-00416-t002:** Different preparations prepared using the BJ-3DP technology, as well as the composition of the powder bed and printing ink used.

	Dosage	Powder Bed		Ink			API and the Parts Where It Exist		Release Behavior	Ref.
Excipients	Binder	Solvent	Binder	Excipients
Rapid release preparations	Instant-dissolving Tablets	MCC, mannitol, colloidal silicon dioxide	PVP K30	Isopropanol aqueous solution	PVP K30	Glycerin	Levetiracetam (65%)	Powder	Dispersion (<30 s) and drug release (2.5 min > 90%)	[52]
	Fast disintegration tablet, low-dose preparations	MCC	PVP	Water-ethanol	PVP	Polysorbate, sodium lauryl sulfate	Quinapril Hydrochloride (2%)	Powder	Drug release (30 min > 80%)	[32]
	Fast disintegration Tablet (hydrophobic API)	MCC, spray dried PVP (with clotrimazole)	PVP	Water-ethanol	PVP	Polysorbate, sodium lauryl sulfate	Clotrimazole (11%)	Powder	Drug release (30 min > 80%)	
	Fast disintegration	Lactose monohydrate	Kollidon VA64	Water	Kollidon VA64	Red liquid food dye	Indomethacin (10%)	Powder	Disintegration time (<10 s)	[53]
	Dispersive tablets	Lactose monohydrate, spray-dried lactose monohydrate, MCC, mannitol, silica	PVP K25	Water-ethanol	Polyethylene glycol 1500	-	Ketoprofen (20%)	Powder	Disintegration time (<25 s)	[54]
Slow-release and controlled-release preparations	Tablets with near zero order release	Kollidon SR, HPMC	-	Drug containing area: water	PVP K17	Tween 20	Pseudoephedrine hydrochloride (50%)	Ink	Near constant release rate, 100% release at 8,12,16 h for different formulations	[55]
				Drug-free area: 75% ethanol and 25% water	Triethyl citrate					
	Zero-level release preparations	HPMC E50, colloidal silicon dioxide	PVP K30	Aqueous 90% ethanol	Ethyl cellulose	-	Acetaminophen (80%)	Powder	98% of drugs released linearly in 12 h	[56]
	Zero-level release preparations	Lactose, HPMC E50	PVP K30	Aqueous 75% ethanol	PVP K30	Glycerol	Diclofenac sodium	Ink	98% of drugs released linearly in 12 h	[57]
Compound preparations	Compound dispersible tablets with multichamber structure	MCC, mannitol, colloidal silicon dioxide	PVP K30	Isopropanol aqueous solution	PVP K30	Glycerin	Levetiracetam (65%) in powder; pyridoxine hydrochloride (4.5%) in ink		Two drug release (both 5 min = 100%)	[58]
	Multi-drug combinations with compartments for compounding	calcium sulfate hemihydrate	-	Aqueous 5% ethanol	-	Tween 80	-		90% lisinopril released in 24 h, above 60% spironolactone released in 24 h	[59]
				Poly (ethylene glycol) diacrylate, PEG200, ethanol			Lisinopril (40 mg/mL) in ink			
							Spironolactone (20 mg/mL) in ink			
Scaffold	Bone scaffold	β-tricalcium phosphate (β-TCP), Fe_2_O_3_, SiO_2_	-	-			-		-	[60]
	Bone scaffold	β-TCP, SiO_2_, ZnO	-	-			-		-	[61]
	Bone scaffold	β-TCP, MgO, ZnO	-	-			-		-	[62]
	Bone scaffold	Hydroxyapatite microsphere	-	Water	PVP, polyvinyl alcohol, polyacryl amide		-		-	[63]
	Biodegradable composite scaffold	Calcium sulfate hemihydrate	2-pyrrolidone	-			-		-	[64]

## 4. Principle of FDM Technology and Applications in the Pharmaceuticals

### 4.1. The Principle of FDM Technology

FDM technology is widely used in pharmaceuticals due to the advantages of simple equipment, low cost, and high product strength. Using computer-aided design software, 3D-printed products are manufactured by depositing molten material layer by layer on printing platforms [65]. The principle is shown in Figure 2. The polymer filament containing the drug is extruded by two rollers through a high-temperature nozzle, and the print head moves in the X-Y axis direction under the control of computer software to print the product; after completing one layer of printing, the printing platform drops or the Z axis rises a distance equal to one layer thickness to start the next layer of printing, and repeats the process until completion [66,67].

Currently, there are three main ways to prepare 3D tablets using FDM technology, as shown in Figure 2B–D: (1) Dipping-melting method: The filament is dipped into a solution or dispersion containing API to obtain a filament containing API for printing. (2) HME-FDM: The API is added to the conveyor together with the excipients, and the filaments containing the API are obtained through the extrusion unit and used for the preparation of 3D-printed drugs, which is currently the most common method. (3) Filling and forming method: first print an empty shell, fill in the API, and then continue to print the shell; the printing and filling processes can be carried out simultaneously or sequentially [65,68,69].

In the HME—FDM method, for example, the drug is first mixed with excipients such as polymers in a molten state, and a filament of the target diameter is extruded at a certain pressure, speed, and shape [65]. The filament is then passed to the heating zone without bending or extrusion, heated to a temperature slightly above its melting point, and then extruded through a nozzle for the preparation of 3D printed tablets. The method requires a high level of mechanical strength and elasticity of the filament containing the drug to avoid breaking or fracturing of the filament during the printing process, thus, affecting the printing accuracy and product quality, and therefore, this technique has major restrictions on the choice of API and printable excipients [70,71,72].

The physicochemical properties of the filament (such as mechanical, thermal, and rheological properties) determine its printability [73]. The polymers used in FDM technology must be thermoplastic and characterized, and the most popular materials used are acrylonitrile butadiene styrene, polylactic acid, polyamide, and polycarbonate [74]. In addition, it has been found that polyvinyl alcohol, a biodegradable material usually used as a support material, has the potential to be upgraded to an important filament material for personalized medicine because it can be dissolved into a colloidal solution [75]. Polymeric materials used in FDM are usually characterized by parameters such as glass transition temperature (T_g_) and melting temperature (T_m_) [76,77]. In particular, the T_g_ of the polymer must be as far away from its degradation temperature as possible [78]. Mackay et al. estimated that T_g_ + 78 °C is the lowest FDM printing temperature for amorphous polymers, a value derived from the average of the lowest FDM printing temperatures for three polymers [72,79]. Secondly, the rheological characteristics of the filament material are also important [80]. Viscosity, as an important characterization of rheological properties, affects not only the ability of the filament to pass through the nozzle at the printing temperature, but also the ability of the filament to recover its structure after deposition [81]. The shear viscosity of the filament through the nozzle depends not only on the internal factors of the material, such as the formulation of the filament, the molecular weight of the drug, and the solid form, but also on external factors, such as the extrusion temperature and shear rate, the nozzle diameter (which is mostly very narrow), and the influence of the printing speed [82]. In addition, homogeneity of the filament, such as the absence of lumps or air bubbles, is essential to achieve layer-by-layer deposition of the filament without thickness variations [83].

### 4.2. FDM Technology in Pharmaceuticals

Based on the properties of FDM technology, it has been widely used to prepare a variety of preparations. For example, the first study to test patient acceptability of 3D-printed preparations of different shapes and sizes was conducted using this technology, while researchers also prepared and tested different colored preparations using it, as shown in Figure 3A,B, which showed that smaller, circular shaped tablets were more acceptable [84]. In addition, Jamróz et al. used FDM technology to prepare aripiprazole oral dispersion films, and in vitro study experiments showed the 3D-printed films had higher dissolution rates. FDM technology can also be used for the preparation of controlled release formulations, for example, Lim et al. used FDM technology to print hollow scaffolds for a constant release of drugs, which consisted of a hollow base covered with a lid, with multiple small holes in the scaffold. It was shown that the scaffolds with holes on the sides showed zero-order kinetics with slow-release properties [66]. FDM printers can be equipped with multiple nozzles to print compounded preparations containing different materials. In 2015, Goyanes et al. used this technology to prepare multilayer capsules and double capsules using paracetamol and caffeine as the model drugs, as shown in Figure 3C, where one layer consists of one drug and the next layer consists of another drug, while the double capsules consist of one drug encapsulated in a shell composed of the other drug [85]. FDM technology has also been used in the field of transdermal drug delivery, with Goyanes et al. [86] in 2016, using FDM technology to create customized nasal masks for patients to better treat acne through topical drug delivery, followed by Muwaffak et al. [87] printing customized wound dressings in the shape of the nose and ear through FDM technology.

## 5. Principle of SSE Technology and Applications in the Pharmaceutical Industry

### 5.1. The Principle of SSE Technology

SSE is an additive manufacturing technology that deposits semisolid material layer by layer, where the extrusion head moves and extrudes the semisolid material in a set trajectory, stacking layers on top of each other until the product is printed [88]. The technology is based on FDM, with the difference that the print material used in this technology is in a semisolid form at room temperature, so the temperature should be controlled when heating to avoid too much material softening due to high temperatures and not being able to retain its shape during deposition. When printing, the print material is contained in a special syringe, and its extrusion can be driven by pneumatic pressure, mechanical energy, or an electromagnetic system, as shown in Figure 4 [27].

The pneumatic-based extrusion systems use compressed air to extrude semisolid materials and are suitable for low- and high-viscosity materials. Mechanically based extrusion systems apply mechanical forces directly to the syringe and are classified as piston or screw types. Compared to pneumatic systems, this extrusion system does not require an air compressor, is simpler and more affordable, and is easier to transport [89]. In addition, the syringes can be changed more easily during the printing process, speeding up the printing efficiency. The electromagnetic drive system opens a valve located at the bottom of the syringe through electrical pulses. The electromagnetic drive system opens a valve located at the bottom of the syringe through electrical pulses. This system is suitable for low viscosity bio-inks with ionic or UV irradiation cross-linking mechanisms and is not suitable for materials with high viscosity [27].

The main difference between SSE and other material extrusion technologies (such as FDM, direct powder extrusion) is the material [88]. The technology uses semisolid or semi-molten materials, which require a high level of material stability to prevent material precipitation or phase separation due to changes in temperature during the printing process, resulting in blocked nozzles and print failure. The rheological properties of the material are particularly important in determining printability and print results [90]. An analysis of the rheological properties of the material based on the printing process is presented below, as shown in Figure 5.

Firstly, the semisolid material should exhibit a high viscosity at rest, a decrease in viscosity, and a certain degree of fluidity after passing through the nozzle with reasonable shear, and the material should also have the ability to recover its viscosity quickly after shear to avoid further flow after deposition onto the platform [91]. Among other things, a decrease in viscosity and rapid recovery after shear require a thixotropic shear-thinning behavior of the semisolid material. This fast recovery property can be achieved by a proper balance between the loss modulus (G”, which determines the flow properties), and the storage modulus (G’, which determines the elasticity and immediate recovery). Secondly, the diameter of the extruded filament should match the size of the nozzle, which is directly related to the balance between G” and G’. If G’ is too high, the elastic stresses stored through the nozzle may be released at the nozzle, causing the filament to swell. If the viscous component of the filament predominates, due to the re-entanglement of the component after shearing, the viscosity of the extruded filament may increase and lead to shrinkage of the filament, making it smaller than the nozzle diameter. For the time being, it is still a challenge to predict the expansion/contraction of nozzle-extruded filaments [92,93]. Thirdly, the filaments should have self-supporting properties when used as an internal filling for the tablets. Among other things, it is required that the filament behave as a spring that hardly deforms under gravity, which requires a certain G’.

After printing, postprocessing processes such as cooling or drying depend on the printing temperature or the use of solvents; by means of cooling, evaporation of solvents, and so on, the final product is obtained [94]. In addition, the type of printhead, print speed, printing temperature, extrusion pressure, etc., are also important for the quality of the final product [27].

### 5.2. SSE Technology in Pharmaceuticals

When SSE technology was used in the field of pharmaceutics, the initial research was also a validation of the feasibility of preparing complex drugs. In 2015, Khaled et al. [95] used SSE technology to manufacture a multi-active tablet containing three drugs, as shown in Figure 6A, with a sustained release zone in the upper layer with compartments separating nifedipine and glipizide, and a captopril osmotic pump in the lower layer, with a connecting layer in the middle that disintegrates rapidly within one hour, allowing the tablet to separate into two parts, resulting in different release effects during dissolution. This study successfully demonstrated that SSE technology can be used for the preparation of complex multicompartment 3D-printed tablets. Shortly afterward, Khaled et al. [94] manufactured a new compound preparation, shown in Figure 6B, which had an upper immediate release zone consisting of acetylsalicylic acid and hydrochlorothiazide and a bottom extended-release zone containing atenolol, ramipril, and pravastatin. The five drugs are separated from each other in the tablet to avoid drug incompatibility problems, and each drug achieves the desired release behavior.

Compared to FDM technology, SSE technology has a lower printing temperature and uses a semisolid material that is well suited for preparing chewable tablets, such as soft candy, greatly enhancing compliance for pediatric patients. In addition, the technology makes it easy to change or replenish consumables simply by changing disposable syringes, and the small size of the device makes it well suited for hospital use. Therefore, research in recent years has tended towards personalized chewable tablets for pediatric patients.

In 2019, Alvaro Goyanes et al. [23] prepared chewable isoleucine tablets for pediatric patients using SSE technology in a hospital setting, as shown in Figure 6C. Six different flavors and colors were prepared, each with different specifications, which were well accepted by pediatric patients, and after taking the tablets, isoleucine levels in the patients’ blood were well controlled. The study was the first to use 3D-printing technology in a clinical setting, advancing the development of personalized medicines. In 2020, Herrada-Manchón et al. [96] prepared chewable tablets of ranitidine hydrochloride using the SSE technique, as shown in Figure 6D, with an interesting and appealing appearance for children in the form of hearts, bears, and other interesting shapes, and the preparation met all requirements for quality homogeneity, dose accuracy, and solubility. In 2021, Tatsuaki Tagami et al. [97] prepared chewable tablets of lamotrigine for pediatric patients using hydrogel materials, as shown in Figure 6E. The preparation was favored by pediatric patients, and the study provides an effective solution for future drug use in pediatric patients in a clinical setting. In 2022, using amlodipine besylate as a model drug, Han et al. [98] developed six different sizes of tablets for use in pediatric patients aged 2–16 years using SSE technology, effectively meeting the clinical need for individualized doses of amlodipine besylate in pediatric patients. Shortly afterward, Zhu et al. [99] from this group prepared chewable tablets based on a gelatin matrix using propranolol hydrochloride as a model drug, with tablet doses that could be flexibly adjusted by the shape and size of the model.

There are also studies of other types of oral preparations. First are the immediate-release preparations, with the advantage of customized dosage. For example, in children with epilepsy, where the dose of levetiracetam varies with the course of treatment, SSE technology makes it easy to obtain a specific dose of levetiracetam tablets according to the needs of the child by changing the size and layers of the model [100,101]. Secondly, the addition of sustained-release materials to semisolid materials allows for the preparation of sustained-release tablets. In 2020, Cheng et al. [102] prepared theophylline tablets using HPMC K4M and E4M as sustained-release materials and examined the optimal HPMC ratio, with the in vitro dissolution results of drug release from the tablets within 12 h. Finally, another class of dosage form is the orally dispersible film (ODF). Sjöholm et al. [103] used the SSE technique to prepare an ODF using warfarin, which has a narrow therapeutic window, as a model drug, solving both the problems of individualized dosing and swallowing difficulties.

There is also some research on medical devices, but it is still at the dosage form exploration stage, validating that SSE technology can be used to prepare these complex medical devices. In 2020, Wu et al. [104] used SSE technology to print a substrate and a cylindrical array of the patch, then stretched the array through a glass plate stretching device to form a needle-like tip. As shown in Figure 6F, the microneedles contained insulin and were able to penetrate the skin of mice to alleviate symptoms in diabetic mice. In 2020, Andriotis et al. [105] used a pectin-based biodegradable ink to print films that could be used directly on wounds or dried as patches using an extruded bio-3D printer. Figure 6G shows the deviation of the dried patch from the theoretical size, which showed that 3D-printed patches could accelerate wound healing through in vitro wound healing studies. In 2021, Seoane-Viano et al. [106] prepared tacrolimus suppositories using SSE technology, as shown in Figure 6H, which were able to release 80% of the drug within 120 min, effectively increasing the concentration of the drug at the site of action and reducing systemic side effects.

**Figure 6 pharmaceutics-15-00416-f006:**
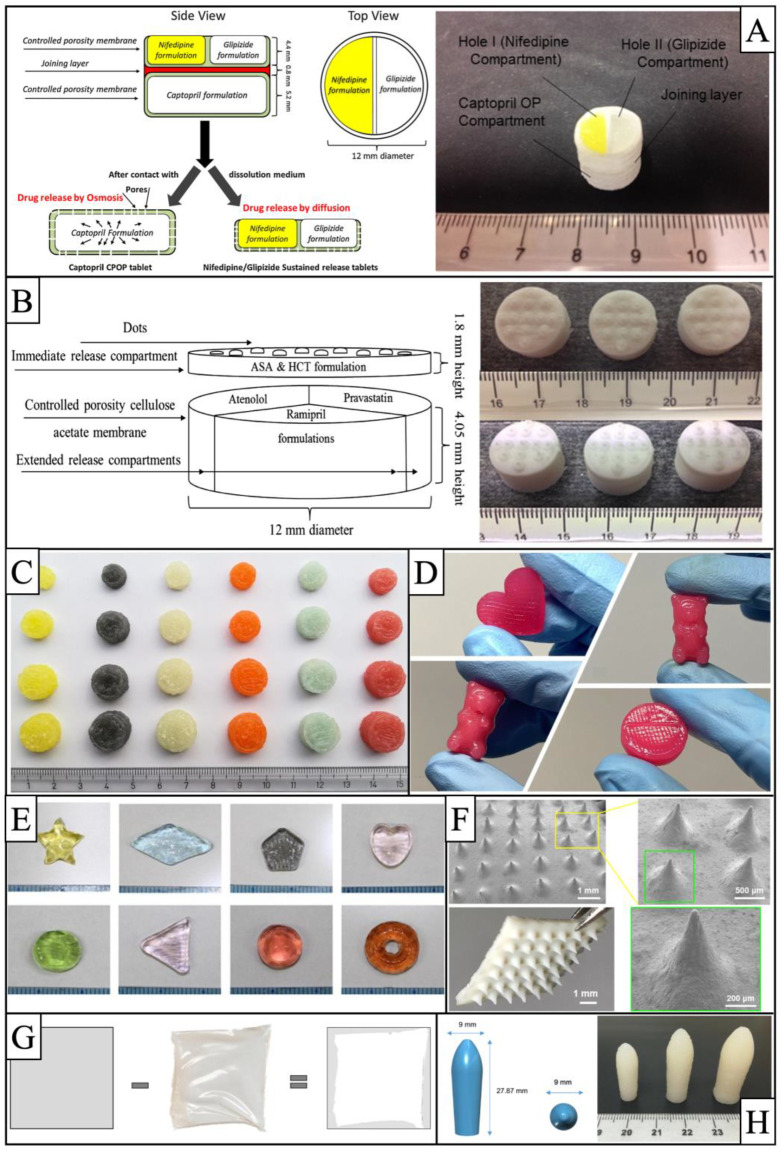
Images of various types of preparations prepared by SSE technology: (**A**) Multi-active tablet containing three APIs with sustained release zones in the upper layer and osmotic pumps in the lower layer, with the model picture on the left and the printed tablet picture on the right [95]; (**B**) compounded tablet containing five APIs, with the model picture on the left, with the immediate-release layer at the top and the extended-release layer at the bottom, and the printed tablets on the right [94]; (**C**) chewable isoleucine tablets with different flavors, colors, and sizes [23]; (**D**) 3D-printed gummies with different shapes [96]; (**E**) 3D-printed gummies with different shapes and colors [97]; (**F**) image of a 3D-printed microneedle, and SEM image of partial enlargement [104]; (**G**) 3D-printed films of the patch, image of dried dimensions versus theoretical dimensions [105]; (**H**) 3D-printed tacrolimus suppositories, with the model image on the left and the different sizes of suppositories on the right [106]. Figures were reproduced and modified with permission from [23,94,95,96,97,104,105,106].

## 6. Principle of MED Technology and Applications in the Pharmaceutical Industry

### 6.1. The Principle of MED Technology

MED 3D printing is a material extrusion technology developed by Triastek that combines both hot melt extrusion and fused deposition modeling technologies. The principle is shown in Figure 7 [26]. It consists of feed and mixing modules, material delivery modules, and multiple print stations, each of which prints one material.

To start with, API and different excipients are added to different feeding devices, which are heated and highly sheared by the hot melt extrusion system to form a homogeneous mixed molten state of the material, which is further transported to the hot melt extrusion module. Finally, the printing stations coordinate with each other, and the different types of materials in the molten state are combined with each other and deposited layer by layer on the printing platform under precise pressure and temperature control in order to obtain 3D-printed preparations of the target structure [26].

Compared to FDM, SSE, or other material extrusion 3D-printing technologies, the MED technology process has no need to prepare filaments or semisolid materials in advance, and there is no secondary heating of materials. The technology allows for the mixing, melting, delivery, and printing of API and excipients in one step, enabling continuous feeding and product production. MED technology is currently considered to be the most useful and clinically relevant 3D-printing technology for solid dosage forms.

### 6.2. MED Technology in Pharmaceuticals

Triastek has taken advantage of the unique advantages of MED technology to develop a range of tablets with different release behaviors by designing drug compartments, multiple drug combinations, using pH-responsive sustained release materials, and varying the surface area of the drug layer or sustained release layer, as described in the article by Yu et al. in 2021 [26]. In this paper, one of the design ideas is chosen, as shown in Figure 8. The pale white region is the outer layer without drugs and impermeable to water, and the bluish region is the inner layer with drugs. The amount of drug released per unit of time is controlled by changing the number of layers and the inner surface area of the inner layer.

## 7. Principle of SLA Technology and Applications in the Pharmaceutical Industry

### 7.1. The Principle of SLA Technology

SLA technology is based on the principle of photopolymerization and uses laser scanning to harden liquid resin to manufacture 3D-printed objects layer by layer [107]. It works as shown in Figure 9. Depending on the printer setup, printing can be performed from the top down or the bottom up. When printing, the liquid tank is filled with liquid photopolymer resin, and the laser beam is focused on the resin surface by the scanning mirror, forming a light spot. The area swept by the light spot will be cured. When a layer of scanning is completed, the printing platform drops one layer in height, the squeegee smooths the resin surface for the next layer of printing, and so on until it is complete. Then, the product is removed, and the excess resin and support structure on it are removed [108]. The technique has a high degree of precision, which is directly related to the spot diameter, with the smallest spot diameter currently at 0.011–0.075 mm and the smallest single layer thickness at 0.01–0.02 mm, making it very suitable for the preparation of microneedles, scaffolds, or other types of medical devices [109].

A photon is usually emitted during the printing process to initiate polymerization, which is the synthesis of a polymer in a chain reaction that requires at least three components: a light source, a monomer/oligomer that can be photopolymerized and a PI [110]. During the polymerization process, the PI reacts in the presence of light and produces initiating substances (such as free radicals, anions, cations, etc.), which can attack and add other monomers/oligomers, and cross-linking takes place. The light curing process can therefore also be divided into photo-induced polymerization and photo-crosslinking [111]. The former refers to a chain reaction in which monomers are added linearly, while the latter refers to a process in which crosslinks are formed between two macromolecular chains [109].

The degree of curing increases with increasing light intensity. According to Jiang et al. [112], the degree of curing increased from 3.1% to 87.7% as the light intensity increased from 5 mW/cm^2^ to 40 mW/cm^2^. They also studied the effect of exposure time on the degree of curing. At the same light intensity, the degree of curing increased from 26.85% to 70.98% when the exposure time was increased from 1.2 s to more than 3 s. When the exposure time was increased from 3 s to 12 s, the degree of curing only increased from 70.98% to 81.74%, indicating that for a fixed amount of resin, light intensity and light time will reach saturation at a certain point.

The rheological properties of the resin also have a strong influence on the SLA process. The viscosity of the resin should ideally be approximately 1 Pas at room temperature to ensure that it is in a liquid state at processing temperature and thus ensure chain flow [83]. The viscosity of the resin can range from 0.1 Pas for low molecular weight polymers to 101 Pas for high molecular weight polymers. For higher viscosity resins, it is possible to process at higher temperatures, but this is limited to formulations that are not sensitive to heat [109,113].

Generally, SLA products require post-curing after printing. The purpose of post-curing is to improve the mechanical properties of the product [114]. Photocentric has several UV light-curing resins for SLA and DLP processes [115]. They recommend a minimum of 2 h of UV exposure time (36 W) for these resins to ensure a high-strength product. Three-dimensional Hubs lists several post-treatment methods for the SLA process, including basic support removal, wet sanding, mineral oil treatment, painting (clear UV-protected acrylic), polishing to transparency, etc. SLA also has some other post-treatment methods, such as surface treatment with sealers, primers, paints, or metal coatings [83,116].

### 7.2. SLA Technology in Pharmaceuticals

SLA technology is more precise than any other 3D printing technology, with a minimum single-layer thickness of 0.01–0.02 mm, but studies of oral preparations have been limited by the paucity of light-cured materials available for oral drug preparation. In 2018, Wang et al. [117] used SLA technology to manufacture acetaminophen and 4-amino salicylic acid extended-release tablets, which reduced drug degradation compared to tablets printed using FDM technology, and this is a clear advantage that provides an alternative route for preparing tablets for heat-sensitive drugs. In 2018, a team of researchers at University College London used SLA technology to manufacture a batch of tablets of different shapes for the study of factors affecting drug release [118], showing that the rate of drug release is largely influenced by the surface-area-to-volume ratio of the tablets, which is of great interest for the preparation of tablets of different doses with equivalent release rates. In addition, the SEM images of the tablets do not show any porosity due to the high precision and high density of cross-linking of the resin through the light curing process, which results in a very tight structure inside the tablets and a slower release of the tablets.

In recent years, SLA technology has been widely used in the field of microneedles due to its high precision and the high strength of the product. Microneedles are micro-scale needles that pass through biological barriers (such as the skin) in a minimally invasive manner, avoiding touching blood vessels and nerves and therefore not causing pain or bleeding. Microneedles are now considered to be a powerful drug delivery system with excellent delivery efficiency. Figure 10 and Table 3 list the relevant literature and characteristics of microneedles prepared by SLA technology in recent years.

## 8. Progress in Commercialization of the 3D Printed Drug Industry

Up to now, the 3D-printed drug industry has been developing for more than two decades. In 1996, the US company Therics received a license for the application of Massachusetts Institute of Technology’s PB 3D-printing technology and founded the world’s first 3D-printed drug company, but due to the great difficulty of development, Therics was ultimately unable to achieve industrialization. In 2003, Aprecia re-licensed the PB technology and, after ten years, developed the ZipDose technology, in 2015 the first 3D-printed drug Spritam^®^ was produced and approved through this technology, setting off a wave of 3D-printed drug research.

Since 2015, the 3D-printed drug industry has entered a period of rapid development, with many professional 3D-printed drug companies emerging one after another. However, the development of 3D-printed drugs requires not only mechanical engineering and pharmacy researchers, but also comprehensive talents in materials science, software, and information engineering, as well as facing strict legal regulations in the pharmaceutical industry, making the overall difficulty high. At present, the global 3D-printed drug industry is still in its infancy, with several pharmaceutical companies accelerating the development and launch of 3D-printed drugs. Figure 11 illustrates the current status of the global 3D-printed drug industry, with companies mainly located in Europe, the US, and China, divided into large-scale production and personalized drug delivery according to the direction of technology application.

### 8.1. Large-Scale Production

The large-scale production is the use of the traditional drug production model, firstly, the development of drug products, followed by the declaration and registration of products, and then the successfully approved drugs will be manufactured by pharmaceutical companies on a large scale and sold around the world. As shown in Figure 11, only the US Aprecia and China’s Triastek have reached the stage of commercialization of drug products, other pharmaceutical companies are still in the exploration stage, such as the US Mercer and Germany’s Merck.

Aprecia was first established in 2003 with the aim of large-scale production, and in 2011, it began operating aGMP-compliant 3D-printed drug production line capable of producing 100,000 tablets per day. Although the launch of Spritam^®^ in 2015 launched a boom in 3D printed drug research, the product’s market reception was mediocre, and Aprecia then transformed itself into a technology-based company, working with pharmaceutical and biological companies to develop or produce drugs. In 2017, Aprecia partnered with orphan drug company Cycle Pharmaceuticals to provide more options for medicines for patients with rare diseases. At the end of 2020, a long-term strategic partnership with Oak Ridge National Laboratory in the US is expected to upgrade the ZipDose 3D printing facility to further expand the application of the technology in the field of 3D printed drugs.

Triastek launched its continuous and intelligent MED 3D-printed drug production line in 2018, from material mixing to tablet formation in one step and controlling quality in real-time through process analysis technology, and has already industrialized MED technology with an annual capacity of 50 million tablets. There are currently four 3D-printed products in the world that have entered registration filings, except for Spritam^®^, the remaining three are from Triastek, two of which have already received INDs from the FDA in the US. Triastek’s business model involves collaborating with other companies in addition to developing pharmaceutical products. Triastek entered a partnership with Siemens in March 2022 to provide global pharmaceutical companies with solutions for digitally developing and manufacturing drugs, and with Eli Lilly in July 2022 to improve drug bioavailability in the gut through precise targeting and programmed drug release.

### 8.2. Personalized Drug Delivery

In addition to production at scale, 3D-printed drug technology is ideally suited for the preparation of personalized drugs for patients with different disease states and different ages. The main application scenario for this model is hospital pharmacies, offering a fast and automated drug option for patients with personalized drug needs. As shown in Figure 11, research in this direction is currently more active in Europe, with major players such as specialist 3D printed drug companies FabRx, Multiply Labs, and DiHeSys, independent research institute TNO, and large multinational drug company AstraZeneca.

The UK company FabRx was founded in 2014 and is currently one of the most active companies in 3D printed drugs, they have researched and explored many 3D printing technologies, including FDM, SLS, SLA, SSE, DPE. With a clear commercial orientation towards personalized drug delivery, the company launched a breakthrough 3D printer in 2020: the M3DIMAKER™, which comprises three technologies: FDM, SSE, and DPE, which allows users to choose the print head according to their needs and allows for fast and flexible preparation of 3D printed drugs based on different principles and materials that can be better used in personalized pharmaceutical scenarios. The company is collaborating with French cancer Centre Gustave Roussy in 2021 to develop personalized drugs for the treatment of early-stage breast cancer.

Multiply labs, founded in 2016 in South San Francisco, US, develops robotic manufacturing platforms to help pharmaceutical companies produce biological medicines. It also develops personalized medicines through a two-step approach, starting with the preparation of capsules of different thicknesses and compartments through FDM technology, and then filling the chambers of the capsules with different drugs or nutrients to enable multi-drug combinations. The company is collaborating with global life sciences company Cytiva, and researchers at the University of California, San Francisco, in 2021 to develop a sophisticated robotic control system that can serve to automate cellular therapies and improve the efficiency of cellular therapy drug production.

## 9. Policies and Regulations in the Field of 3D Printed Drugs

As an emerging technology in the pharmaceutical industry, 3D printing has many advantages, and the 3D-printed drug industry is moving towards modern personalized medicines, which is directly related to the efforts of pioneering companies and the active support of government agencies, such as the Drug Review Centre.

Spritam^®^, the world’s first 3D-printed preparation, received IND approval from the FDA in 2013. Subsequently, to encourage and facilitate the successful approval of new technology products in the pharmaceutical industry, the FDA established the ETT in 2014, and the ETT’s involvement directly ensured the successful approval of Spritam^®^ in 2015. In January 2017, the FDA published a review of a new chapter in pharmaceutical manufacturing: 3D-printed drug products, arguing that 3D printing is an emerging technology for the future; in July of the same year, the FDA issued an industry guidance on the advancement of emerging technology applications for pharmaceutical innovation and modernization, pointing out that 3D-printing technology and continuous manufacturing are important strategic directions. In 2019, the China CDE also published a review, indicating its recognition and concern for the 3D-printed drug industry, and looking forward to the arrival of 3D printing to accelerate the era of personalized and intelligent drug delivery.

In 2020, Triastek’s MED 3D printing technology was applied to join the FDA’s Emerging Technology Program and was approved, representing that the technology was recognized at the regulatory level. In January 2021, T19, the world’s second 3D printing mechanism, received FDA IND approval. In the same year, Triastek also participated in the Q13: Continuous Manufacturing conference organized by CDE in China, helping to drive pharmaceutical technology innovation. In 2021, the National Academies of Sciences, Engineering, and Medicine, at the request of the Center for Drug Evaluation and Research, produced a report on innovations in drug manufacturing that concluded that 3D-printing technology is a completely new manufacturing method compared to traditional methods of drug production and will replace them.

Thus far, no regulatory body has issued guidelines for 3D-printed preparations, and there is an urgent need to establish regulatory standards for them. It is believed that as the technology continues to develop and researchers continue to explore, 3D-printing technology can establish a set of scientific standards in the pharmaceutical field, from theory to practice and from production to regulation.

## 10. Conclusions

This paper reviews the relevant literature on several 3D-printing technologies commonly used in the pharmaceutical industry, elucidating the principles and characteristics of each technology, the dosage forms suitable for each technology, and the development trend; and reporting on the commercialization direction of representative companies or institutions of 3D-printed drugs, their development history, and the breakthrough results achieved, driving the innovation of drug development models. As an emerging technology, the registration and filing path for 3D-printed preparations is unique, while intellectual property rights, drug regulations, and other policies or regulations are still breaking new ground. Overall, this review aims at reflecting the current development status, industrial characteristics, and overall development trends of 3D-printed drugs. We hope that this review can provide a meaningful reference for those who are engaged in related research. It is believed that with continuous efforts, the future of the 3D-printed drug industry is promising and will certainly promote drug preparation technology that is intelligent and personalized.

## Figures and Tables

**Figure 1 pharmaceutics-15-00416-f001:**
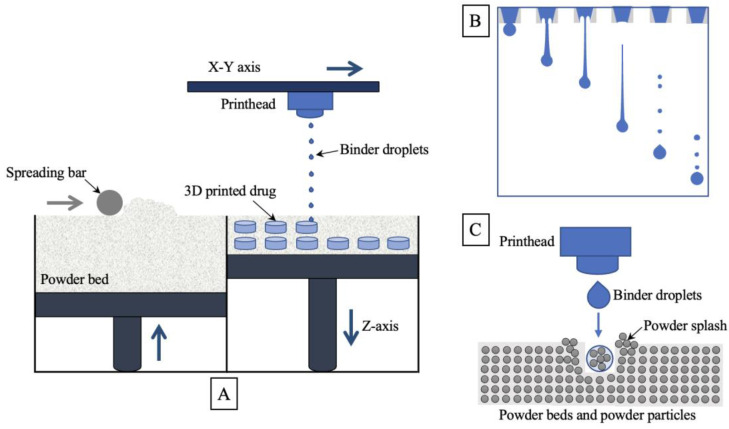
Schematic diagram of the principle and mechanism of tablet preparation by BJ-3DP technology: (**A**) schematic diagram of the printing principle of BJ-3DP technology; (**B**) schematic diagram of the flight state of a droplet after ejection through the nozzle; and (**C**) schematic diagram of droplet impact on the powder bed.

**Figure 2 pharmaceutics-15-00416-f002:**
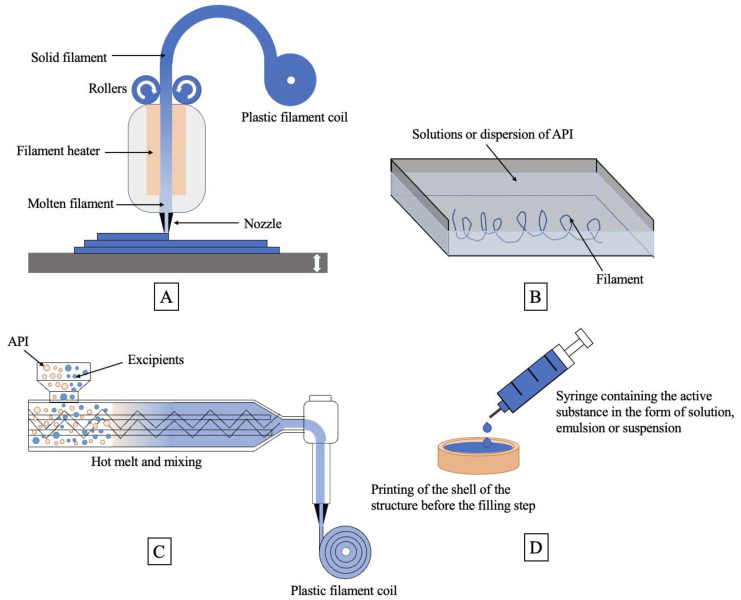
Schematic diagram of the principle of FDM technology and three methods of tablet preparation: (**A**) schematic diagram of the printing principle of FDM technology; (**B**) schematic diagram of the preparation of drug-containing filaments by the dipping-melting method; (**C**) schematic diagram of the preparation of drug-containing filaments by the HME-FDM method; and (**D**) schematic diagram of the preparation of tablets by the filling and forming method.

**Figure 3 pharmaceutics-15-00416-f003:**
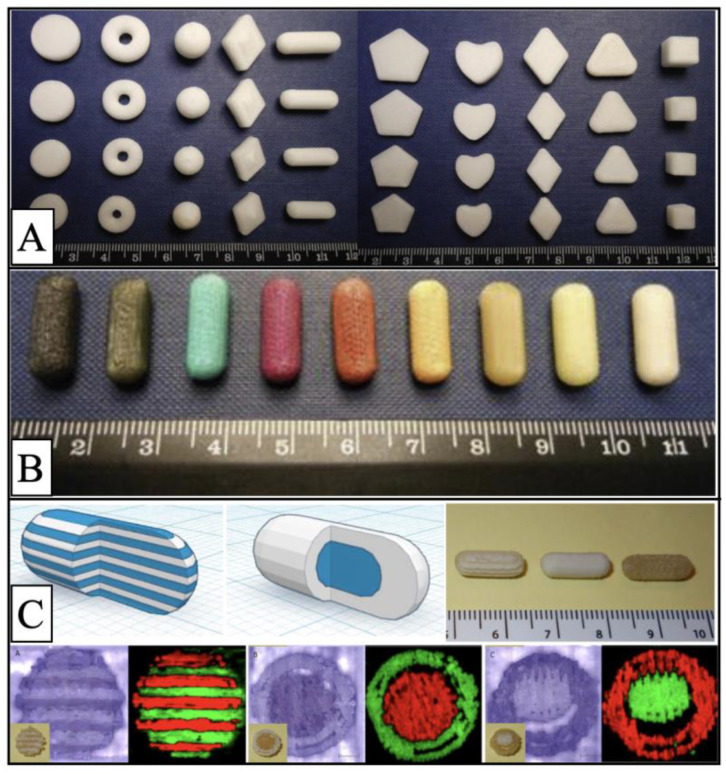
Images of various types of tablets prepared by FDM technology: (**A**) Images of 3D-printed tablets with different shapes and sizes [84]; (**B**) images of 3D-printed tablets in capsule form with different colors [84]; (**C**) sectioned multilayer device and sectioned DuoCaplet (caplet in caplet) model images, 3D-printed preparations, and white light and 2-dimensional Raman mapping images [85]. Figures were reproduced and modified with permission from [84,85].

**Figure 4 pharmaceutics-15-00416-f004:**
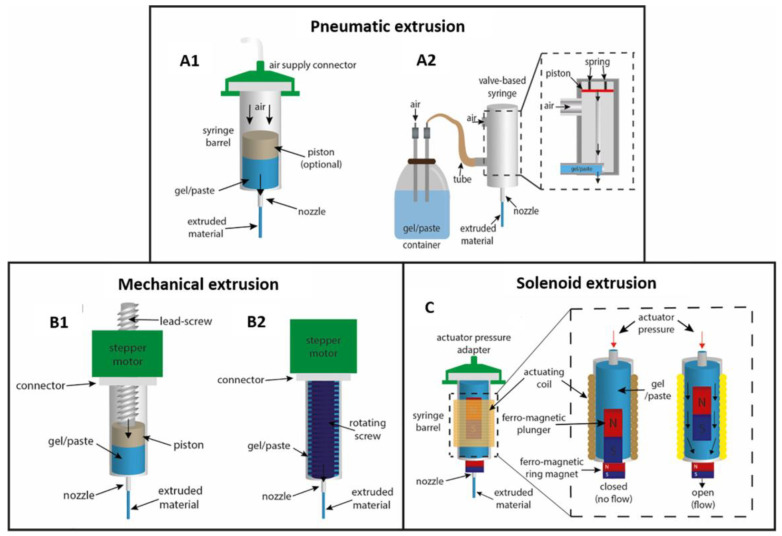
SSE 3DP extrusion mechanisms: (**A**) pneumatic extrusion, including (**A1**) valve-free and (**A2**) valve-based, (**B**) mechanical extrusion, including (**B1**) piston- or (**B2**) screw-driven, and (**C**) solenoid extrusion [27]. Figures were reproduced and modified with permission from [27].

**Figure 5 pharmaceutics-15-00416-f005:**
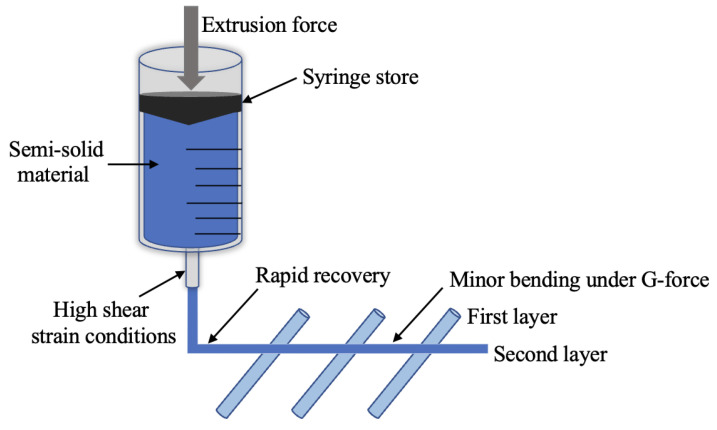
Diagram of semisolid material passing through a nozzle under extrusion and the rheological challenges that should be overcome.

**Figure 7 pharmaceutics-15-00416-f007:**
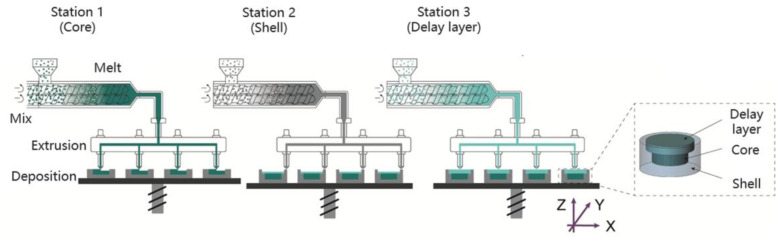
The principle of collaborative preparation of tablets with delayed release shells using multiple print stations through MED technology. Figures were reproduced and modified with permission from [26].

**Figure 8 pharmaceutics-15-00416-f008:**
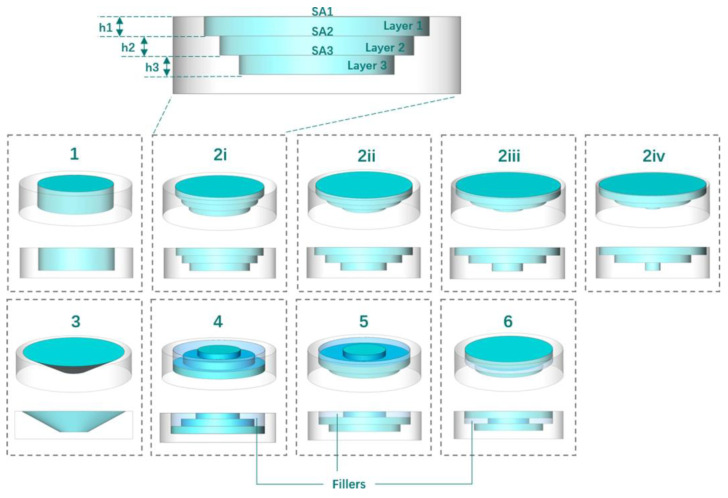
Multilayered drug compartments in core–shell structure tablets with constant or varied SA. Drug compartments (teal), shells (pale white), and fillers (bluish). (1) Drug compartment with constant SA. (2i–2iv) Drug compartment with stepwise decreasing SA. (3) Drug compartment with continuously decreasing SA. (4) Drug compartment with stepwise increasing SA. (5) Drug compartment with increasing–decreasing SA. (6) Drug compartment with decreasing-increasing SA. Figure reproduced and modified with permission from [26].

**Figure 9 pharmaceutics-15-00416-f009:**
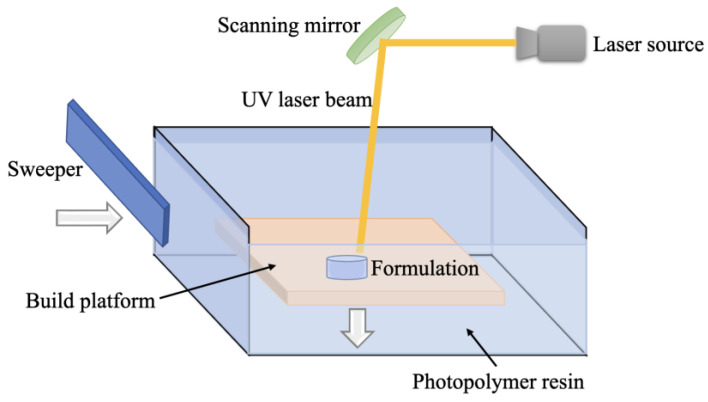
Schematic diagram of the printing principle of SLA technology.

**Figure 10 pharmaceutics-15-00416-f010:**
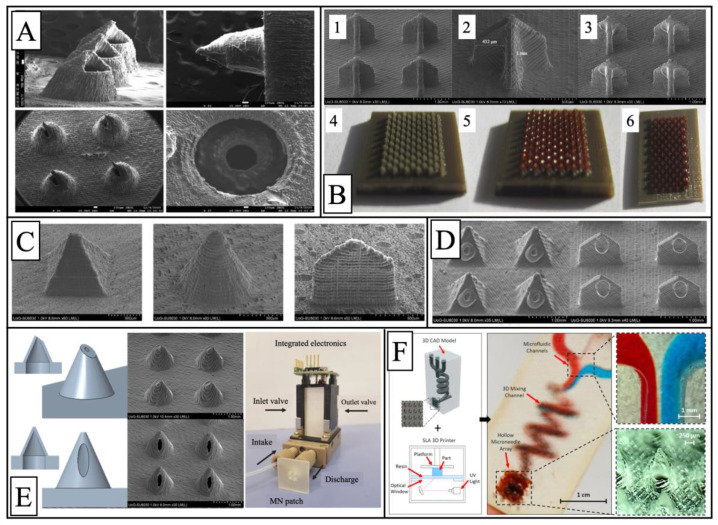
Images of various types of microneedles prepared by SLA technology: (**A**) SEM images of the 3D-printed microneedles at different angles [119]; (**B**) uncoated microneedles physical view (4) and SEM images (1,2), coated microneedles physical view (5,6) and SEM images (3) [120]; (**C**) 3D-printed microneedles in different shapes [121]; (**D**) pyramidal and spear-shaped microneedles with insulin coatings [122]; (**E**) from left to right are the model images of the hollow microneedles, the SEM images, and the 3DMNMEMS configuration [123]; (**F**) 3D-printing of microfluidic-enabled hollow microneedle devices [124]. Figures reproduced and modified with permission from [119,120,121,122,123,124].

**Figure 11 pharmaceutics-15-00416-f011:**
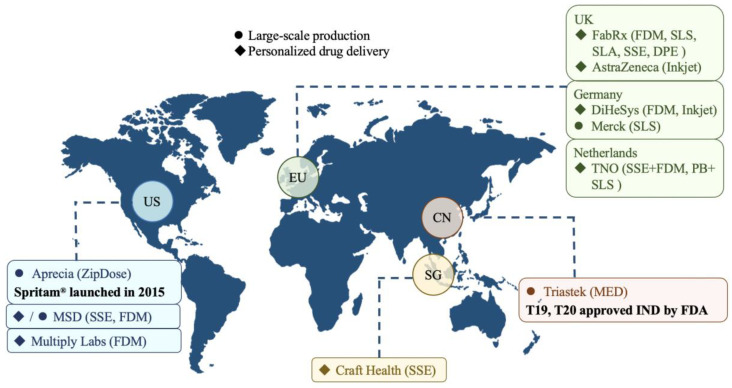
Global industry developments in 3D printed drugs.

**Table 1 pharmaceutics-15-00416-t001:** Characteristics, advantages, and disadvantages of different 3D printing technologies for drug preparation.

Types of 3D Printing Technology		Technical Characteristics			Advantages	Disadvantages
		Preprocessing	Print Processing	Postprocessing		
BJ-3DP		Prefabricated powder bed or ink containing drug	Room temperature/heating	Removal and recovery of powders, drying of preparations	❿ Wide range of available excipients ❿ High drug loading capacity ❿ No support material required ❿ Suitable for immediate release preparations	❿ Lack of flexibility in product design ❿ Complex post-processing ❿ High requirements for packaging and transportation ❿ Larger equipment
Material Extrusion	FDM	Prefabricated filamentous containing drugs	Heating	Removal of support material/none	❿ Simple and inexpensive equipment ❿ Ability to create a variety of 3D structured preparations	❿ Required to prefabricate drug-containing filaments with suitable mechanical properties ❿ Fewer material options available ❿ High printing temperatures ❿ Low drug loading capacity
	SSE	Prefabricated semi-solid materials containing drugs	Room temperature/heating	Drying/none	❿ Simple and inexpensive equipment ❿ Can be printed at room temperature ❿ Use of disposable syringes for easy change of material	❿ Prefabricated semisolid materials with suitable properties ❿ Often requires post-processing ❿ Low printing accuracy
	MED	None	25–250 °C	None	❿ High precision ❿ Wide range of available printing materials ❿ Complete industrial production line available	❿ The process of material mixing and melting to extrusion needs to be monitored in real time ❿ The properties of the materials used need to be well understood in order to set the right parameters
SLA		Prefabricated polymer monomers containing drugs	Photopolymerization	Separation from unreacted polymer monomers and re-curing	❿ High printing accuracy and can be used to prepare microneedles ❿ Can be printed at room temperature	❿ Few print materials available ❿ Long preprocessing process ❿ Postprocessing process available

**Table 3 pharmaceutics-15-00416-t003:** Different types of microneedles prepared using SLA technology.

Microneedle Types	Resin Materials	API	Microneedle Shapes	Research Findings	Ref.
Coated microneedle	Biocompatible Class I resin: Dental SG	Insulin	Pyramid and flat spear shaped	Microneedles prepared by SLA technology penetrate better than metal microneedles; they help to lower glucose levels quickly and maintain them for longer than direct insulin injections.	[122]
Hollow microneedle	Class IIa biocompatible resin which is a mixture of methacrylic acid esters and photo initiator comprised of (in % *w*/*w*) >70% methacrylic oligomer, <20% glycol methacrylate, <5% pentamethyl-piperidyl sebacate, and <5% phosphine oxide	-	Syringe-shaped	The microfluidic microneedle device prepared by SLA technology allows for the homogeneous mixing of multiple fluids at different flow rates for transdermal delivery, making it particularly suitable for preclinical studies of multiple drug treatments.	[124]
Hollow microneedle	Biocompatible class I resin: methacrylic oligomers and phosphine oxides as photo initiators	Insulin	Cone-shaped with top and side openings	Combining 3D printing, microneedles, and microelectromechanical systems, a novel device for multifunctional and drug-controlled transdermal drug delivery, has been successfully prepared and its feasibility has been demonstrated by drug delivery.	[123]
Coated microneedle	Class I biocompatible resin	Cisplatin	Cross-shaped	Demonstrates the potential of 3D-printed microneedles for transdermal delivery of the anticancer drug cisplatin in nude mice, where cisplatin is sufficiently permeable to achieve high anticancer activity and tumor regression.	[120]
Hollow microneedle	Biocompatible Class I resin	Rifampicin	With subapical holes present in a quarter of the needle tip	Microneedles with subacute holes at the tip quarter were designed for transdermal drug delivery of the antibiotic rifampicin, with effective penetration and ideal bioavailability in SD rats.	[119]
Coated microneedle	Biocompatible Class I acrylic resin: Dental SG	Insulin	Cone, pyramid, spear-shaped	The effects of geometry and manufacturing parameters on the quality and performance of microneedles are investigated to optimize the ability of 3D-printed microneedles for drug delivery.	[121]

## Data Availability

Not applicable.

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
