# Peer review of "A Review of 3D Printing Technology in Pharmaceutics: Technology and Applications, Now and Future"

_pharmaceutics, 2023, doi:10.3390/pharmaceutics15020416_

Round 1

Reviewer 1 Report

In this article the authors present a review on 3D printing technologies.  The article is well written with detailed examples and up to date literature reports.  The authors provide an impressive amount of information including technical details for the various 3D printing technologies.

I enjoyed reading the article as it covers a wide range of trends such as personalised dosage forms, scale up and industrial applications of 3D printing.  I am sure it will attract the interest of several researchers in the field.

I do have a minor suggestions for the authors about paediatric applications.  They could add the following articles in the relative section as there is strong interest in this area: 

doi: 10.3390/pharmaceutics13081301

doi: 10.1016/j.ijpharm.2022.122135

doi: 10.1007/s11095-017-2284-2

Author Response

Thank you very much for your careful review of the manuscript and favorable comments!

Thank you for your suggestion, we have added the three references to the appropriate places in section 2.1, with reference numbers 17-19.

Reviewer 2 Report

This manuscript is the introduction of latest and useful information containing pharmaceutical industry of 3D printing (e.g. Triastek). T19 and T20 from Triastek which are tablet formulation prepared by MED 3D printing was also introduced well. They introduced recent key articles for each type of 3D printer. The manuscript also mentioned policy and regulation issue of 3D printed drug. I think this manuscript has good balance, is useful and worth for publishing in this journal. I left small comments.

-3.1. Spritam is swallowable 3D printed tablet composed of levetiracetam which is water soluble drug. ZipDose technology is applicable for all BCS class I-IV drugs. The company may mention the use of solid dispersion and cyclodextrin, although I do not know the appropriated source. If the information is included in this manuscript, the manuscript may be better.

-5.2. Regarding 3D printed medicine prepared SSE printer, other dosage forms are proposed and fabricated by researchers as the authors introduced. If it is possible, can you add more references? -Minitablet, oral mucoadhesive film (buccal film), other patch formulation, redispersible nanomedicine, others…

-7.2 Regarding 3D printed tablet prepared by SLA 3D printer which is synonym of DLP 3D printer, the typical tablet (PEGDA-based tablet) is ghost tablet. These tablets do release drug but do not disintegrate. If it is possible, can you add the information with appropriate reference?

Author Response

Thank you for your hard work and suggestions for the manuscript.

(1) Section 3.1. Thank you for your suggestion, it is helpful to further understand the application of BJ-3DP technology in pharmaceuticals, but there is less relevant research available and we have added it appropriately in section 3.1.

(2) Section 5.2. Thank you for your suggestion, our description of the use of SSE technology for other dosage forms is lacking and we have added the appropriate content and references to section 5.2.

(3) Section 7.2. Thank you for your suggestion, which will further the understanding of the characteristics of SLA technology in oral preparations, we have enriched the description of the internal structure of tablets prepared by researchers at the University of London using SLA technology, which has been added in section 7.2.

Reviewer 3 Report

The authors of the review manuscript focused on the pharmaceutical use of 3D printing. In the manuscript, the authors describe and analyze the most commonly used technologies and the development trend of the pharmaceutical industry toward 3D printing.
Basically, the manuscript is well-structured and logical, and the tables and figures are informative.
Some critical comments:
- It would be helpful for the reader when the manuscript has a list of abbreviations. The list could include both technological methods and excipients mentioned in the manuscript.

-         The filament materials for FDM need more details, e.g. PVA can be widely used since it is dissolved to form a colloidal solution (Basa B. et al: Evaluation of biodegradable PVA-based 3D printed carriers during dissolution. Materials 2021, 14, 1350, DOI:10.3390/ma14061350)
Unfortunately, the resolution of the abbreviation 3DMNMEMS in Figure 10 is missing.
- In the case of Figure 6, it is not indicated that the referees (original authors) have agreed that the authors will use their published figures.
- The third photo in part B of Fig. 10 is only marked as an SEM image. It is not specified whether it is coated, uncoated, or what kind of microneedles are shown in the picture.

Author Response

Thank you for your hard work and suggestions for the manuscript.

1. Thank you for your suggestion, we have added a list of abbreviations to the first page of the manuscript and have trimmed the abbreviations that appear in the body of the manuscript.

2. Thank you for your suggestion, we have added content about FDM filament materials in section 4.1.

3. Thank you for your suggestion, we have fixed the problem with the resolution of the image in Figure 10 and replaced it.

4. Sorry for the confusion, A-F in Figure 6 have been copyrighted to the journal, and G-H are open access.

5. Thank you for your suggestion, we have modified the legend description in Figure 10B.